# Hyperspectral Anomaly Detection Based on Spectral Similarity Variability Feature

**DOI:** 10.3390/s24175664

**Published:** 2024-08-30

**Authors:** Xueyuan Li, Wenjing Shang

**Affiliations:** 1School of Physics and Electronic Information, Yantai University, Yantai 264005, China; shangwenjing@ytu.edu.cn; 2Shandong Yuweng Information Technology Co., Ltd., Yantai 264005, China

**Keywords:** hyperspectral anomaly detection, spectral similar variability feature, feature fusion, deep learning, residual network, autoencoder

## Abstract

In the traditional method for hyperspectral anomaly detection, spectral feature mapping is used to map hyperspectral data to a high-level feature space to make features more easily distinguishable between different features. However, the uncertainty in the mapping direction makes the mapped features ineffective in distinguishing anomalous targets from the background. To address this problem, a hyperspectral anomaly detection algorithm based on the spectral similarity variability feature (SSVF) is proposed. First, the high-dimensional similar neighborhoods are fused into similar features using AE networks, and then the SSVF are obtained using residual autoencoder. Finally, the final detection of SSVF was obtained using Reed and Xiaoli (RX) detectors. Compared with other comparison algorithms with the highest accuracy, the overall detection accuracy (AUC_ODP_) of the SSVFRX algorithm is increased by 0.2106. The experimental results show that SSVF has great advantages in both highlighting anomalous targets and improving separability between different ground objects.

## 1. Introduction

Hyperspectral remote sensing image processing technology is a branch of the signal processing field, many signal processing-related methods provide theoretical and technical support for hyperspectral remote sensing image processing. According to the characteristics of hyperspectral remote sensing images, remarkable achievements have been achieved in the application directions of hyperspectral image classification [1,2,3], unmixing [4,5,6], super-resolution mapping [7,8], and target detection [9]. In recent years, many experts and scholars have systematically reviewed different types of hyperspectral remote sensing image processing methods. samples include hyperspectral spatial enhancement techniques or super resolution (SR) [10], the application of machine learning to lithology mapping and mineral exploration [11], and the application of deep learning to anomaly detection [12]. These systematic reviews provide important reference and guidance for the further research and development of hyperspectral remote sensing image processing, and strongly promote the continuous innovation and improvement of related technologies.

Although hyperspectral images are rich in spectral and spatial information, they still face various challenges in the research process, including redundancy of high dimensional data, pollution of spectral noise and atmospheric influence, mixed pixels, and different objects within the same spectrum and in different spectra of the same object. Spectral dimension transformation involves mapping hyperspectral images to the corresponding feature space through the feature processing method, which makes the ground objects indistinguishable in the original feature space separable in the new feature space. In hyperspectral abnormal target detection, spectral dimension transformation can improve the separability between the background and the anomaly target. The most common feature processing methods are principal component analysis [13] (PCA), independent component analysis [14] (ICA) and nonlinear principal component analysis [15]. The essence of spectral dimension transformation is to obtain higher-level features of original hyperspectral images by a mapping method and improve the accuracy of anomaly detection by using its ability to improve the separability between different ground objects. The hyperspectral anomaly detection (HAD) of differential images [16] utilizes difference images to estimate background changes during the feature extraction stage, so as to suppress background signals and highlight anomaly signals. Fractional Fourier entropy [17] employs fractional Fourier transform for pre-processing, then uses space–frequency representation to obtain features from the intermediate region between the original spectrum and its complementary Fourier transform. Unsupervised spectral mapping and feature selection [18] highlights an anomaly target by searching for the optimal feature subset from the candidate feature space while mapping high-dimensional features to a low-dimensional space using unsupervised neural networks.

In addition, research on HAD that is based on a linear model has also found some success. The linear model is able to obtain the error term of the hyperspectral image. For example, the hyperspectral image is mapped to the feature space of other dimensions, and then the features of the mapping are re-projected to the original feature space by the opposite method. Finally, the reconstruction error is taken as the value of the anomaly degree. Because normal samples are easier to reconstruct than anomaly samples, the samples with higher reconstruction errors are considered abnormal targets. For example, the residuals between the reconstructed image and the original image are obtained by PCA projection reconstruction, and the projection parameters are updated in several iterations [19]. The work of [20] is enhanced by that of [19], and potential anomaly target is filtered out according to the error value of each iteration. The reconstruction probability algorithm of an autoencoder (AE) [21] is also a detection model that can obtain reconstruction errors through feature mapping. The joint graph detection (JGD) [22] model considers both spectral and spatial features. Through the spectral sub-model, the reconstruction error between the original hyperspectral sensing image (HSI) and the feature image after the graph Fourier transform (GFT) is mapped to fractional Fourier entropy (FrFE), which enhances the anomaly detection capability and shows the advantage in distinguishing background anomalies. To solve the problem of the PCA being sensitive to feature scales and outliers, robust PCA (RPCA) [23] decomposes data into low-rank and sparse matrices to enhance robustness to noise and outliers. RPCA integrating sparse and low-rank priors (RPCA-SL) is a new variant that achieves a more precise separation by combining prior targets and is solved using a near-end gradient algorithm. A discriminant reconstruction method based on spectral learning (SLDR) [24], firstly uses a spectral error map (SEM) to detect the anomaly, and then uses a spectral angle distance (SAD) to restrict the AE to follow a unit Gaussian distribution. The obtained SEM can well reflect the spectral similarity between the identification and reconstruction. The mixture of Gaussian low-rank and sparse decomposition [25] decomposes the HSI into low-rank background and sparse components, then deduces the hybrid Gaussian model of sparse components by variable decibels, and finally calculates the anomaly by Manhattan distance. Pixel-associate AE [26] uses super-pixel distance to build two representative dictionaries, and then obtains the hidden layer expression of similarity measure by AE.

In HAD, spectral dimension transformation involves mapping the background and anomaly target of hyperspectral data to another feature space, so that it can then identify the background and anomaly target that cannot be separated in the original feature space and thus improve the detection accuracy. However, the traditional feature mapping method is to map both the background and anomaly target to the same feature space; however, this cannot effectively highlight the anomaly target. The main factor affecting this problem is the uncertainty of the mapping direction. It is difficult to separate the anomaly target from the background effectively by the conventional spectral dimension transformation. 

To solve this problem, hyperspectral anomaly detection based on the spectral similarity variability feature (SSVF) was proposed. First, the AE network is used to fuse high-dimensional similar neighborhoods into lower-dimensional similar features, which have similar information to neighborhood pixels and can reduce the computational burden of subsequent networks. The SSVF is then obtained using an autoencoder of the residuals, essentially to acquire the error between the image itself and its similar neighbors. Finally, the RX detector was used to obtain the final detection result of SSVF. Therefore, the proposed algorithm is also called SSVFRX. Hyperspectral images in which the background and its neighboring image elements have a high degree of similarity can be initially judged to belong to the same ground feature. In contrast, anomalous targets and their similar pixels have a low probability of belonging to the same ground feature. The similarity change feature allows most of the same features to be mapped in the same direction, while anomaly targets are mapped in the opposite direction.

This paper evaluates the superiority of SSVF in improving the difference between anomaly target and background through experiments. Comparative experiments are used to judge the effect of introducing a similar neighborhood on the improvement of the difference between the abnormal target and the background. SSVF aims to solve the uncertainty of the mapping direction of the spectral dimension transformation method for an unsupervised network model by introducing a similarity difference value to obtain a mapping direction which can improve the difference between anomaly target and background.

## 2. Materials and Methods

### 2.1. Experiment Data Description

The superiority of the SSVFRX algorithm was verified using seven hyperspectral experimental datasets. The detailed parameters of the experimental datasets are shown in Table 1, and the false-color images and their ground truth images are shown in Figure 1. Additionally, the following must be explained:(1)D_1_ and D_2_ are from the Remote Sensing and Image Processing Group (RSIPG) repository [27], captured at an altitude of 1200 m on a sunny day. D_1_ is the full image, while D_2_ is a cropped portion containing an anomaly. Both datasets have undergone residual stripe removal, and D_1_ has been further processed with noise whitening and partial spectral discarding.(2)D_3_ and D_4_ are from the San Diego Airport, with the anomaly target being aircraft.(3)D_5_ is from the Digital Imaging and Remote Sensing (DIRS) laboratory, which is part of the Chester F. Carlson Center for Imaging Science at the Rochester Institute of Technology.(4)The high-spectral datasets D_6_ and D_7_ are from the personal website of Xudong Kang, School of Electrical and Information Engineering, Hunan University. The original images were downloaded from the AVIRIS website [28]. The authors extracted 100 × 100 sub-images and applied a noise level estimation method to remove the noisy bands.

**Figure 1 sensors-24-05664-f001:**
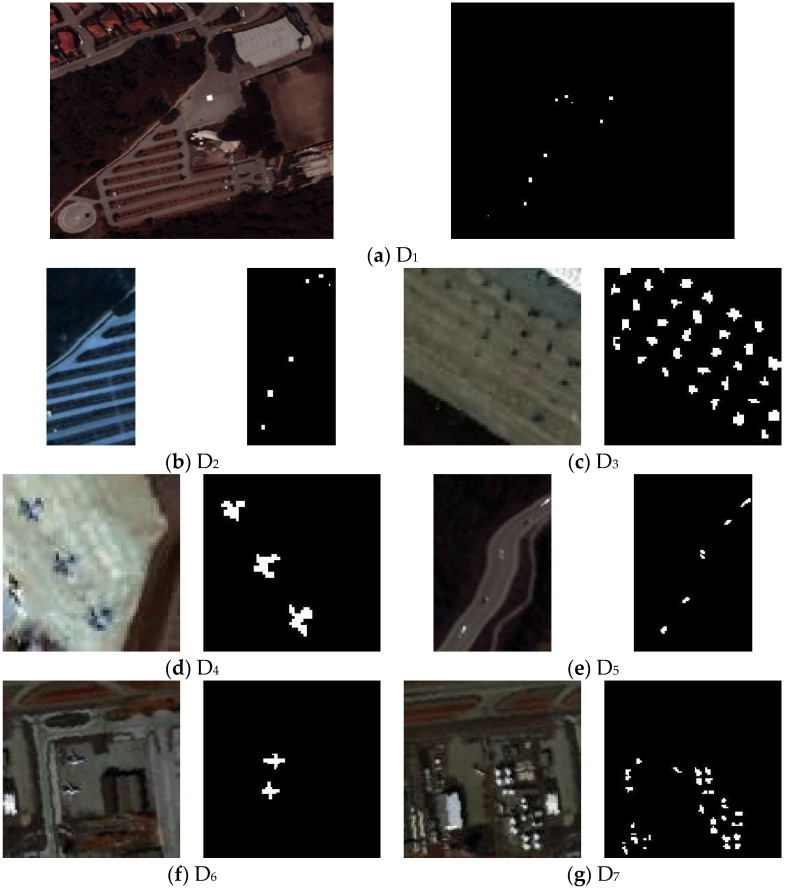
False-color image and target position of experimental data.

**Table 1 sensors-24-05664-t001:** Experimental dataset parameter.

Data Set Name	Hyperspectral Imaging Sensor	Collected Location	Spectral Range	Spectral Resolution	Spatial Resolution	Size of Origin Image	Size of Sub-Image	The Original Number of Bands	Number of Bands after Processing
			μm	nm	(*m*)	Pixel	Pixel		
D_1_	VNIR-SIM.GA	Parking lot in suburban vegetated	0.40–1.00	1.2	0.6	375 × 450	375 × 450	511	127
D_2_							200 × 100	511	511
D_3_	AVIRIS	San Diego	0.36~2.50	9.0	3.0	400 × 400	80 × 80	224	126
D_4_							60 × 60		
D_5_	ProSpecTIR-VS2 sensor	Avon, NY.	0.39~2.45	5.0	1.0	--	120 × 80	360	360
D_6_	AVIRIS	Los Angeles	0.36–2.50	9.0	7.1	100 × 100	100 × 100	224	205
D_7_									

### 2.2. Hyperspectral Anomaly Detection Based on Spectral Similar Variability Feature

The proposed algorithm is divided into three main steps: data pre-processing, similar feature fusion (SFF) and spectral similarity variability feature extraction. The overall flow chart is shown in Figure 2. The process of data pre-processing involves processing the origin HSI by PCA and whitening. SFF refers to the fusion of similar features from multiple similar neighborhoods, using AE networks to obtain a low-dimensional feature representation of the same dimension as the original image. Spectral similarity change feature extraction refers to the calculation of the difference value between similar features and the original features using a residual autoencoder network. Finally, the final detection result is obtained by the RX detector.

#### 2.2.1. Data Pre-Processing

In hyperspectral image processing, the pre-processing stage is crucial for improving data quality and subsequent analysis effectiveness. Before network training, the hyperspectral images are usually pre-processed, such as by reducing dimension and whitening.

The hyperspectral dataset is represented as $X=\left\{x^{\left(1\right)},x^{\left(2\right)},...,x^{\left(N\right)}\right\}$, where $x^{\left(i\right)}=\left\{x_{1}^{\left(i\right)},x_{2}^{\left(i\right)},...,x_{n}^{\left(i\right)}\right\},x_{j}^{\left(i\right)}$ is the $j^{\mathrm{th}}$ dimension of the $i^{\mathrm{th}}$ sample, N is the number of samples, and n is the sample dimension.

The data pre-processing process is shown in Figure 3. Firstly, principal component analysis (PCA) is used to obtain the feature after reducing dimension $X_{p}$, and then whitening is used to obtain the whitened features $X_{w}$.
(1)$$X_{w}=X_{p}/\sqrt{\lambda_{i}}={\left(\Sigma_{k}\right)}^{T}X/\sqrt{\lambda_{i}}$$
where $\Sigma =\frac{1}{N}\sum_{i=1}^{N}x^{\left(i\right)}{\left(x^{\left(i\right)}\right)}^{T}$ is the covariance matrix, $\lambda_{i}$ is the $i^{\mathrm{th}}$ eigenvalue of the covariance matrix, and $\Sigma_{k}$ is the first *k* columns of the covariance matrix. 

#### 2.2.2. Similar Feature Fusion Based on Autoencoder

Hyperspectral images have strong high-dimensional properties, and their similar features can be reconstructed in a manner that is nearly lossless by an autoencoder for images with similar features. This process can help to further improve the separability between classes through its own nonlinear transformations while reducing the training burden of the subsequent residual network. The fusion model of similar features fusion is shown in Figure 4.

The Euclidean distance is used as a similarity measure to find the nearest neighboring sample in the sample set. The feature of each sample is represented as $x^{\left(i\right)}$ and its neighborhood is represented as $S^{\left(i\right)}=\left\{S_{1}^{\left(i\right)},S_{2}^{\left(i\right)},...,S_{Q}^{\left(i\right)}\right\}$, also known as the set of Q neighborhoods nearest to $x^{\left(i\right)}$ in the dataset, where Euclidean distance is used as a similarity measure. The specific process is as follows:

First, calculate the similarity, as follows:(2)$$d_{i}=\overset{N}{\underset{j=1}{⋂}}{\left\|x^{\left(i\right)}\;-\;x^{\left(j\right)}\right\|}^{2}$$
where $d_{i}$ the similarity set of the $i^{\mathrm{th}}$ sample.

Then, the similarity matrix $d_{i}$ is arranged from small to large, and the first Q samples are selected, as its similarity neighborhood set is $S^{\left(i\right)}=\left\{S_{1}^{\left(i\right)},S_{2}^{\left(i\right)},...,S_{Q}^{\left(i\right)}\right\}$.

The autoencoder is used to undertake similar feature fusion. The training sample set can be represented as $S=\left\{S^{\left(1\right)},S^{\left(2\right)},...,S^{\left(N\right)}\right\}$. The network structure is shown in Figure 3. The network uses gradient descent to minimize the objective function, as follows:(3)$$J(\alpha ,\beta )=\left[\frac{1}{M}\sum_{i=1}^{M}J(\alpha ,\beta ;S^{\left(i\right)},S^{\left(i\right)})\right]+\frac{\lambda }{2}\sum_{l=1}^{n_{l}-1}\sum_{i=1}^{s_{l}}\sum_{j=1}^{s_{l+1}}{\left(\alpha_{\mathrm{ji}}^{\left(l\right)}\right)}^{2}$$
where α, β is the network parameter, M is the number of batches, $S^{\left(i\right)}$ is the $i^{\mathrm{th}}$ input similar sample, λ is the weight decay term, $n_{l}$ is the number of layers of the network, and $s_{l}$ is the number of nodes at layer l.

After the training, the fixed parameters and the expression of the hidden layer are obtained.
(4)Y=f(α×S)

Finally, a nonlinear similar feature Y, which contains similar information is obtained.

#### 2.2.3. Spectral Similar Variability Feature

Although hyperspectral images are very rich in spectral information, because of illumination, noise and other factors, the spectral information of pixel exists in the phenomenon of ‘same object and different spectrum’ (that is, the spectrum of the same object is different). This difference is defined as spectral variation (SV), and the extracted spectral variation information is called the spectral variation feature. The questions of how to extract the spectral variation information and how to use it to enhance the performance of anomaly detection are the highlights of the study in this section.

Every pixel has its similar pixel in the global scope, so that the spectral features combined with other similar pixels are called similar features (SF). Suppose similar pixels belong to different spectra of the same category, the variation between them is called the spectral similar variation feature (SSVF). The advantages of SSVF in hyperspectral anomaly detection may lie in the following aspects. First, the different characteristics of the background and anomaly target show that there is a large variation between the anomaly target as outliers and their similar features. Second, it can be seen from the different spectral changes of different ground object types that SSVF can distinguish different ground object types in scenes to a large extent.

#### 2.2.4. Spectral Similar Variability Feature Extraction Based on Residual Autoencoder

In the similar feature fusion stage, a similar fusion feature **Y** with similar information of multiple neighboring pixels is obtained by using the autoencoder. In order to obtain the variability feature between the SFF and the original features, the residual autoencoder network is used to take the SFF as inputs and the original features as labels. The structure of the residual autoencoder network is shown in Figure 5, and the method of obtaining the SSVF is as follows:

First, the activation value of the network is obtained by forwarding propagation, as follows:(5)$$Z=f_{2}({\theta_{2}\times f}_{1}\left(\theta_{1}\times Y\right))+Y$$
where $f_{1}\left(x\right)=\frac{1}{1+e^{-x}}$, $f_{2}\left(x\right)=x$, $\theta_{1}$ and $\theta_{2}$ are parameters of the network, and Y is a similar feature. 

The purpose of the residual autoencoder is to obtain the error generated when samples in the similar feature space are mapped to the original feature space. The parameter $\theta_{1},\theta_{2}$ is adjusted through back-propagation to minimize the cost function J (the mean square error of the sample set).
(6)$$J\left(\theta_{1},\theta_{2}\right)=\frac{1}{2}{\left\|Z^{\left(i\right)}-X^{\left(i\right)}\right\|}^{2}$$
where $Z^{\left(i\right)}$ represents the i^th^ activation values and $X^{\left(i\right)}$ represents the i^th^ original hyperspectral data.

Then the difference between the activation value Z and the input Y of the residual network is used as the error for back-propagation to update the network parameters $\theta_{1}$ and $\theta_{2}$.

After the training is completed, the spectral similar variability feature can be obtained as follows:(7)E=Z− X

The detection results are obtained by the following methods:(8)$$R\left(E\right)=\mathrm{RXdetector}\left(E\right)$$
where $\mathrm{RXdetector}\left(\cdot \right)$ represents the RX anomaly detection algorithm and $R\left(E\right)$ represents the detection results of the feature sets.

**Figure 5 sensors-24-05664-f005:**
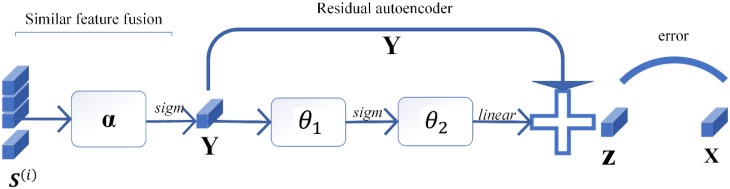
SSVF extraction model.

## 3. Experimental Result

### 3.1. Comparison Algorithm

In this experiment, 10 groups of related comparison algorithms were selected to verify the superiority of the SSVFRX algorithm. Global RX detector (GRXD) [29] is the most basic detection method in the field of anomaly target detection and is widely used in a variety of anomaly detection fields. GRXD, based on PCA [13], is the most commonly used feature extraction method. Principal component reconstruction error (PCRE) [19] is the anomaly detection method based on the residual (error) caused by PCA projection in the reconstruction of original images. Anomaly detection based on autoencoder (ADAE) [21] is a method used to detect an anomaly target through the residual of the autoencoder. Hyperspectral anomaly detection by fractional Fourier entropy (FrFE) [17] is an anomaly detection method based on feature extraction and selection. The low-rank and sparse decomposition model with a mixture of Gaussian (LSDMMoG) [25] is an anomaly detection method for constructing hybrid Gaussian models based on sparse components and low-rank backgrounds. Information entropy estimation based on point-set topology (IEEPST) [30] combines point-set topology and information entropy theory to reveal data characteristics and data arrangement in topological space. Hyperspectral anomaly detection based on chessboard topology (CTAD) [31] refers to the use of checkerboard topology to mine high-dimensional data features. Hyperspectral anomaly detection with guided autoencoder (GAED) [32] is a guided multi-layer autoencoder that reduces the feature representation of the anomaly target by providing feedback. 

### 3.2. Parameter Selection

In order to better generalize the model, the parameter selection phase focuses on selecting a common hyperparameter that applies to most of the data. Therefore, it mainly explains how to adjust the parameters within a certain range.

(1)The first parameter to be adjusted is Q (the number of K neighbors). Because the number of K neighbors directly affects the dimension of input data in the phase of similar feature fusion, the value of Q should not be too large in order for it not to affect the computational efficiency. Take D_3_ as an example, as shown in Table 2, when Q = 9, the anomaly detection accuracy reaches its maximum. However, if Q = 9, then, when the data set dimension is 511, the input data dimension will be as high as 4599, which will affect the computational efficiency of the algorithm. Therefore, Q is set to 5 at this stage.

(2)In order to ensure the stability of detection results, when the network reaches the convergence state, the error of detection performance is small. The hyperparameters can be adjusted to control the degree and speed of network convergence and avoid falling into a local optimum in the following ways:

The main parameters are learning rate (a), learning rate decay (b), maximum number of iterations (T) and batch size. The first of these is used to control the attenuation speed. According to experience, a = 0.1. As the number of iterations increases, a(t) = b × a(t − 1). However, through debugging, the algorithm convergence speed is slow when b ≠ 1, and it is easy to fall into a local optimum, so b = 1. Batch size refers to the sample size of a model training process. It is related to the number of training samples, and a small sample may only need one batch of training. Although large batch size can improve the training speed, it may also cause slow convergence, low generalization performance and even over-fitting. If the batch size is small, data need to be loaded more frequently. Experience has shown that batch size is usually 1% of the sample size (batch size = N × 1%). The number of iterations, T, depends on the convergence degree and speed after the above parameters are determined and is generally set to 100 times according to the convergence situation.

(3)n_0_ is the implicit layer dimension of the residual autoencoder. The mapping direction of the hyperspectral image is controlled by adjusting n_0_. Different mapping spaces affect the separability of different features. Based on experience, this is usually set to n − 20, where n is the original data dimension.(4)n_1_ is the dimension of the last layer of the residual network. As the algorithm needs to obtain the difference between similar fusion features and the original data, it must be consistent with the original image dimension.

### 3.3. Experimental Results

The basic evaluation indexes adopted in this chapter mainly include the three-dimensional receiver operating characteristic (3D ROC) [33], statistical separability analysis (SSA) [34] and detection result image (DRI) [35]. Seven experimental data and five comparison algorithms were selected to verify the superiority of SSVFRX.

The 3D ROC is an extension of the traditional ROC curve, where the threshold τ is used as an independent variable to illustrate the three-dimensional relationship among PD, PF, and τ. Here, P_D_ represents the probability of correctly identifying a target when the true value is indeed a target, also known as the probability of detection. P_F_ represents the probability of incorrectly identifying a target when the true value is a non-target, also known as the probability of false alarm. Figure 6, Figure 7, Figure 8, Figure 9, Figure 10, Figure 11 and Figure 12 display the 3D ROC curves for seven sets of experimental data, along with their corresponding 2D projections. Based on these, new quantitative performance metrics is redefined. AUC_(D,F)_ is the area under the P_D_ and P_F_ curve, while AUC_(D,τ)_ is the area under the P_D_ and τ curve. Both of these metrics are positively correlated with target detection performance, meaning that the higher the value, the better the detection performance. AUC_(F,τ)_ is the area under the P_F_ and τ curve and is negatively correlated with background suppression performance, meaning that the lower the value, the better the background suppression. In addition, several comprehensive indicators are defined below:

AUC_TD_ = AUC_(D,F)_ + AUC_(D,τ)_, which represents target detectability (TD). AUC_BS_ = AUC_(D,F)_ − AUC_(F,τ)_, which represents background suppressibility (BS). AUC_SNPR_ = AUC_(D,τ)_/AUC_(F,τ),_ which measures the signal-to-noise ratio by treating the target as the signal and the background as noise. AUC_TDBS_ = AUC_(D,τ)_ − AUC_(F,τ)_, which represents TD within the background. AUC_ODP_ = AUC_(D,τ)_ + (1 − AUC_(F,τ)_), which represents overall detection accuracy.

The aforementioned metrics across the 7 experimental datasets are presented in Table 3, Table 4, Table 5, Table 6, Table 7, Table 8 and Table 9, where ↑ indicates that the value is proportional to the performance, ↓ indicates that the value is inversely proportional to the performance, and bold indicates the optimal solution. After analyzing the AUC results for these datasets, the following results are obtained:(1)**Background suppressibility (BS)**: AUC_(F,τ)_ and AUC_BS_ correlate with BS capacity.

The SSVFRX model exhibits a number of characteristics in experiments on BS. In most experimental datasets, SSVFRX has the best AUCBS (comprehensive BS) performance. Despite the low performance of AUC_(F,τ)_ under a single hypothesis, its comprehensive BS is strong. In addition to this, in datasets D_4_, D_5_, the AUCBS of SSVFRX are second only to one model, which is a different model, and the difference is very small, 0.0013 and 0.0185, respectively, which suggests that SSVFRX has a superior performance in background suppression.

**Table 3 sensors-24-05664-t003:** AUC performance comparison of different methods on D_1_.

D_1_	AUC_(D,F)_↑	AUC_(D,τ)_↑	AUC_(F,τ)_↓	AUC_TD_↑	AUC_BS_↑	AUC_SNPR_↑	AUC_TDBS_↑	AUC_ODP_↑
GRXD	0.8688	0.0903	0.0148	0.9591	0.8540	6.1026	0.0755	0.9443
PCA	0.8794	0.0912	0.0127	0.9706	0.8667	7.1904	0.0785	0.9579
PCRE	0.7000	0.1924	0.1131	0.8924	0.5869	1.7016	0.0793	0.7793
ADAE	0.6972	0.1779	0.0134	0.8750	0.6838	13.3162	0.1645	0.8617
FrFE	0.8506	0.0846	0.0132	0.9352	0.8373	6.3933	0.0714	0.9219
LSDMMoG	0.8164	0.1960	0.0725	**1.0124**	0.7440	2.7039	0.1235	0.9399
IEEPST	0.6724	0.0529	**0.0002**	0.7253	0.6722	**321.4102**	0.0527	0.7251
CTAD	0.6146	0.1481	0.0043	0.7627	0.6103	34.4880	0.1438	0.7584
GAED	0.7070	**0.2073**	0.0410	0.9143	0.6660	5.0568	**0.1663**	0.8733
SSVFRX	**0.8826**	0.0992	0.0076	0.9818	**0.8750**	13.1049	0.0917	**0.9743**

**Figure 6 sensors-24-05664-f006:**
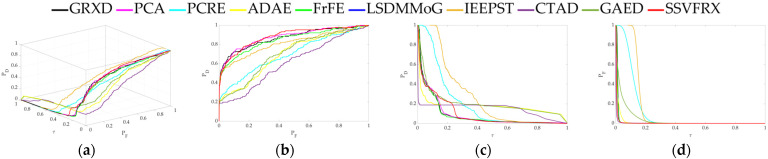
Performance comparison of the 3D ROC of different methods on D_1_. (**a**) Three-dimensional ROC curves. (**b**) Corresponding 2D ROC curves (P_D_,P_F_). (**c**) Corresponding 2D ROC curves (P_D_,τ). (**d**) Corresponding 2D ROC curves (P_F_,τ).

**Table 4 sensors-24-05664-t004:** AUC performance comparison of different methods on D_2_.

D_2_	AUC_(D,F)_↑	AUC_(D,τ)_↑	AUC_(F,τ)_↓	AUC_TD_↑	AUC_BS_↑	AUC_SNPR_↑	AUC_TDBS_↑	AUC_ODP_↑
GRXD	0.5667	0.1394	0.0900	0.7061	0.4768	1.5490	0.0494	0.6161
PCA	0.5714	0.1326	0.0820	0.7039	0.4894	1.6173	0.0506	0.6220
PCRE	0.6666	0.0573	0.0087	0.7240	0.6580	6.6090	0.0487	0.7153
ADAE	0.7674	0.0082	0.0014	0.7756	0.7660	**6.0602**	0.0069	0.7743
FrFE	0.5903	0.0971	0.0513	0.6874	0.5389	1.8906	0.0457	0.6360
LSDMMoG	0.6461	**0.1630**	0.0931	0.8091	0.5530	1.7513	0.0699	0.7160
IEEPST	0.6519	0.0009	**0.0009**	0.6528	0.6510	1.0352	0.0000	0.6520
CTAD	0.5694	0.0372	0.0380	0.6066	0.5314	0.9796	-0.0008	0.5686
GAED	0.6520	0.0284	0.0057	0.6804	0.6463	4.9857	0.0227	0.6747
SSVFRX	**0.8725**	0.1556	0.0432	**1.0281**	**0.8293**	3.6058	**0.1125**	**0.9849**

**Figure 7 sensors-24-05664-f007:**
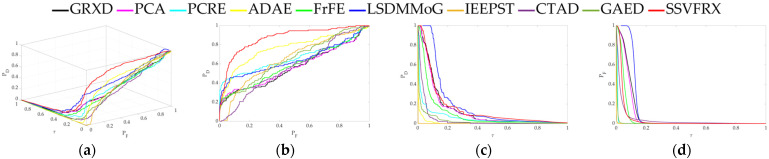
Performance comparison of the 3D ROC of different methods on D_2_. (**a**) Three-dimensional ROC curves. (**b**) Corresponding 2D ROC curves (P_D_,P_F_). (**c**) Corresponding 2D ROC curves (P_D_,τ). (**d**) Corresponding 2D ROC curves (P_F_,τ).

**Table 5 sensors-24-05664-t005:** AUC performance comparison of different methods on D_3_.

D_3_	AUC_(D,F)_↑	AUC_(D,τ)_↑	AUC_(F,τ)_↓	AUC_TD_↑	AUC_BS_↑	AUC_SNPR_↑	AUC_TDBS_↑	AUC_ODP_↑
GRXD	0.8227	0.0858	0.0555	0.9086	0.7673	1.5466	0.0303	0.8531
PCA	0.8170	0.0882	0.0574	0.9052	0.7596	1.5355	0.0307	0.8478
PCRE	0.7546	0.0137	**0.0101**	0.7684	0.7446	1.3665	0.0037	0.7583
ADAE	0.7855	0.0238	0.0143	0.8092	0.7711	1.6623	0.0095	0.7949
FrFE	0.6637	0.3039	0.2909	0.9676	0.3727	1.0445	0.0129	0.6766
LSDMMoG	0.7368	**0.4303**	0.3719	**1.1671**	0.3649	1.1571	0.0584	0.7952
IEEPST	0.7141	0.0332	0.0122	0.7473	0.7019	2.7111	0.0210	0.7351
CTAD	0.8079	0.1436	0.0467	0.9515	0.7612	3.0775	**0.0970**	0.9049
GAED	0.7122	0.0379	0.0258	0.7502	0.6864	1.4692	0.0121	0.7244
SSVFRX	**0.9495**	0.0665	0.0148	1.0161	**0.9347**	**4.4901**	0.0517	**1.0012**

**Figure 8 sensors-24-05664-f008:**
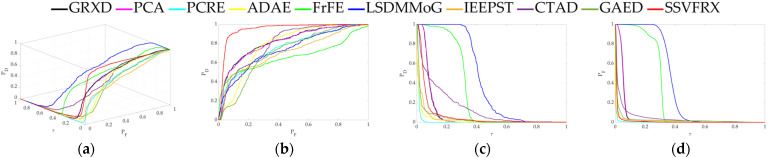
Performance comparison of the 3D ROC of different methods on D_3_. (**a**) Three-dimensional ROC curves. (**b**) Corresponding 2D ROC curves (P_D_,P_F_). (**c**) Corresponding 2D ROC curves (P_D_,τ). (**d**) Corresponding 2D ROC curves (P_F_,τ).

**Table 6 sensors-24-05664-t006:** AUC performance comparison of different methods on D_4_.

D_4_	AUC_(D,F)_↑	AUC_(D,τ)_↑	AUC_(F,τ)_↓	AUC_TD_↑	AUC_BS_↑	AUC_SNPR_↑	AUC_TDBS_↑	AUC_ODP_↑
GRXD	0.8139	0.0539	0.0296	0.8678	0.7842	1.8206	0.0243	0.8382
PCA	0.8168	0.0651	0.0369	0.8819	0.7799	1.7657	0.0283	0.8450
PCRE	0.7127	0.0321	0.0242	0.7448	0.6885	1.3285	0.0079	0.7207
ADAE	0.9417	0.0662	**0.0101**	1.0080	**0.9316**	6.5373	0.0561	0.9979
FrFE	0.9237	0.2895	0.0556	1.2132	0.8680	5.2037	0.2339	1.1575
LSDMMoG	0.7801	0.4411	0.3899	1.2213	0.3902	1.1313	0.0512	0.8313
IEEPST	0.8726	0.0666	0.0149	0.9392	0.8578	4.4808	0.0517	0.9243
CTAD	0.9335	**0.4497**	0.1232	**1.3832**	0.8103	3.6514	**0.3266**	**1.2600**
GAED	0.9048	0.2292	0.0123	1.1340	0.8925	**18.6209**	0.2169	1.1217
SSVFRX	**0.9653**	0.2759	0.0350	1.2412	0.9303	7.8855	0.2409	1.2062

**Figure 9 sensors-24-05664-f009:**
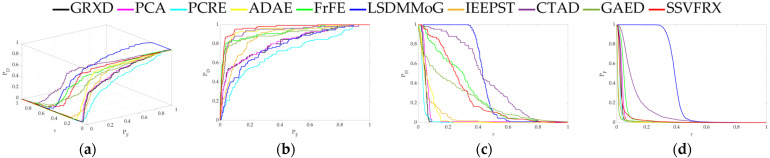
Performance comparison of the 3D ROC of different methods on D_4_. (**a**) Three-dimensional ROC curves. (**b**) Corresponding 2D ROC curves (P_D_,P_F_). (**c**) Corresponding 2D ROC curves (P_D_,τ). (**d**) Corresponding 2D ROC curves (P_F_,τ).

**Table 7 sensors-24-05664-t007:** AUC performance comparison of different methods on D_5_.

D_5_	AUC_(D,F)_↑	AUC_(D,τ)_↑	AUC_(F,τ)_↓	AUC_TD_↑	AUC_BS_↑	AUC_SNPR_↑	AUC_TDBS_↑	AUC_ODP_↑
GRXD	0.9332	0.3309	0.0989	1.2641	0.8342	3.3450	0.2320	1.1652
PCA	0.9675	0.1913	0.0099	1.1589	0.9576	19.2935	0.1814	1.1489
PCRE	0.9652	0.1572	0.0088	1.1223	0.9564	17.9522	0.1484	1.1136
ADAE	0.9703	0.1076	0.0054	1.0779	0.9650	20.1104	0.1022	1.0726
FrFE	0.8675	0.3510	0.1238	1.2185	0.7437	2.8349	0.2272	1.0947
LSDMMoG	0.9309	0.2925	0.0781	1.2235	0.8528	3.7434	0.2144	1.1453
IEEPST	0.9885	0.2305	**0.0024**	1.2190	**0.9861**	**96.8195**	0.2281	1.2167
CTAD	0.9907	**0.5718**	0.0571	**1.5625**	0.9336	10.0140	**0.5147**	**1.5054**
GAED	0.9512	0.1424	0.0083	1.0936	0.9428	17.1093	0.1341	1.0852
SSVFRX	**0.9968**	0.3703	0.0292	1.3670	0.9676	12.6855	0.3411	1.3379

(2)**Target detectability (TB)**: AUC_(D,F)_, AUC_(D,τ)_, AUC_TD_ and AUC_TDBS_ represent the TD in different cases.

Combining the detection results in Table 3, Table 4, Table 5, Table 6 and Table 7, the SSVFRX model has the best AUC_(D,F)_ performance among all experimental data. However, SSVFRX generally performs worse in the AUC_(D,τ)_ of a single hypothesis. This may be due to limitations in the target detection ability under different threshold conditions.

The AUC_TD_ of SSVFRX is ranked 2nd, 1st, 2nd, 2nd, 2nd, 2nd, 1st, 3rd in D_1_~D_7_, respectively, which indicates that the target detection performance is relatively stable in different scenarios and performs well in most of the cases. The AUC_TDBS_ of SSVFRX is ranked 5th, 1st, 3rd, 2nd, 2nd, 1st, 2nd in D1~D7, respectively. This indicates that in terms of the ability of TD to remove BS, SSVFRX performs relatively consistently and excels in most cases.

**Figure 10 sensors-24-05664-f010:**
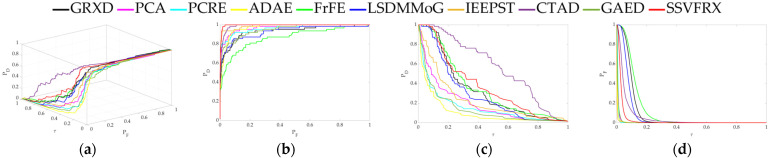
Performance comparison of the 3D ROC of different methods on D_5_. (**a**) Three-dimensional ROC curves. (**b**) Corresponding 2D ROC curves (P_D_,P_F_). (**c**) Corresponding 2D ROC curves (P_D_,τ). (**d**) Corresponding 2D ROC curves (P_F_,τ).

**Table 8 sensors-24-05664-t008:** AUC performance comparison of different methods on D_6_.

D_6_	AUC_(D,F)_↑	AUC_(D,τ)_↑	AUC_(F,τ)_↓	AUC_TD_↑	AUC_BS_↑	AUC_SNPR_↑	AUC_TDBS_↑	AUC_ODP_↑
GRXD	0.8404	0.1841	0.0516	1.0245	0.7888	3.5691	0.1325	0.9729
PCA	0.9278	0.0988	0.0133	1.0266	0.9145	7.4488	0.0855	1.0133
PCRE	0.9348	0.1103	0.0091	1.0451	0.9257	**12.1097**	0.1012	1.0360
ADAE	0.8940	0.0531	0.0128	0.9471	0.8812	4.1437	0.0403	0.9343
FrFE	0.9441	0.1433	0.0241	1.0875	0.9200	5.9377	0.1192	1.0633
LSDMMoG	0.8420	**0.2989**	0.0946	1.1409	0.7474	3.1600	0.2043	1.0463
IEEPST	0.7970	0.0028	**0.0012**	0.7998	0.7959	2.3848	0.0016	0.7987
CTAD	0.7914	0.1991	0.0503	0.9905	0.7411	3.9597	0.1488	0.9402
GAED	0.8745	0.1209	0.0341	0.9954	0.8404	3.5434	0.0868	0.9613
SSVFRX	**0.9767**	0.2470	0.0227	**1.2238**	**0.9541**	10.9006	**0.2244**	**1.2011**

**Figure 11 sensors-24-05664-f011:**
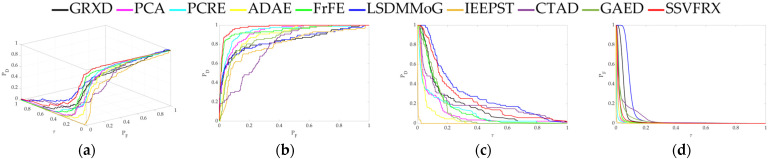
Performance comparison of the 3D ROC of different methods on D_6_. (**a**) Three-dimensional ROC curves. (**b**) Corresponding 2D ROC curves (P_D_,P_F_). (**c**) Corresponding 2D ROC curves (P_D_,τ). (**d**) Corresponding 2D ROC curves (P_F_,τ).

(3)**Overall detection accuracy**: AUC_ODP_ represents the overall detection accuracy.

The overall detection results show that the SSVFRX model has higher AUC_ODP_ scores in most of the datasets. This reveals that it has an advantage in overall detection accuracy. It should be noted in particular that AUC_ODP_ only differed by 0.054, 0.1675 and 0.1374 compared with CTAD in D_4_, D_5_ and D_7_, respectively, but even so, the performance of SSVFRX is still the best model besides CTAD. This shows that SSVFRX has a better overall detection performance than other global detection methods and outperforms local detection methods on most datasets.

SSA is used to assess the separability of the anomaly target and the background. The red box indicates the range of values for the anomaly target and the green box indicates the range of values for the background. The distance between the lower limit of the red box and the upper limit of the corresponding green box reflects the degree of separability between the anomaly target and the background. A larger distance represents a higher degree of separability between the anomaly target and the background, or, in other words, a more prominent anomaly target. The height of the green box represents the degree of background suppression, and the smaller the height, the higher the degree of background suppression. As shown in Figure 13, SSVFRX can significantly improve the separability between the background and the anomaly target and suppress the background. In particular, in datasets D_4_, D_5_ and D_7_, SSVFRX has a lower degree of separability than CTAD, but a higher degree of background suppression.

**Table 9 sensors-24-05664-t009:** AUC performance comparison of different methods on D_7_.

D_7_	AUC_(D,F)_↑	AUC_(D,τ)_↑	AUC_(F,τ)_↓	AUC_TD_↑	AUC_BS_↑	AUC_SNPR_↑	AUC_TDBS_↑	AUC_ODP_↑
GRXD	0.9692	0.1461	0.0437	1.1153	0.9255	3.3406	0.1024	1.0716
PCA	0.9672	0.1170	0.0320	1.0842	0.9352	3.6581	0.0850	1.0522
PCRE	0.9645	0.1315	0.0390	1.0960	0.9255	3.3686	0.0924	1.0569
ADAE	0.9016	0.1080	0.0166	1.0096	0.8850	6.5042	0.0914	0.9930
FrFE	0.9663	0.1168	0.0281	1.0831	0.9382	4.1516	0.0887	1.0550
LSDMMoG	0.9509	0.3805	0.1843	1.3314	0.7665	2.0644	0.1962	1.1471
IEEPST	0.8584	0.0239	**0.0017**	0.8822	0.8567	**14.2165**	0.0222	0.8806
CTAD	0.9575	**0.4095**	0.0424	**1.3670**	0.9152	9.6661	**0.3671**	**1.3246**
GAED	0.8129	0.0865	0.0360	0.8994	0.7769	2.4027	0.0505	0.8634
SSVFRX	**0.9775**	0.2322	0.0224	1.2096	**0.9550**	10.3460	0.2097	1.1872

**Figure 12 sensors-24-05664-f012:**
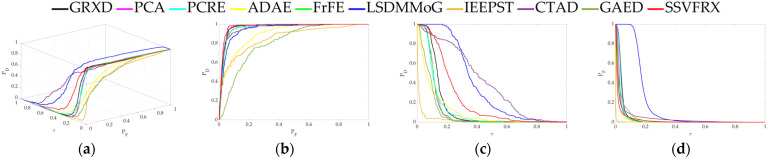
Performance comparison of the 3D ROC of different methods on D_7_. (**a**) Three-dimensional ROC curves. (**b**) Corresponding 2D ROC curves (P_D_,P_F_). (**c**) Corresponding 2D ROC curves (P_D_,τ). (**d**) Corresponding 2D ROC curves (P_F_,τ).

**Figure 13 sensors-24-05664-f013:**
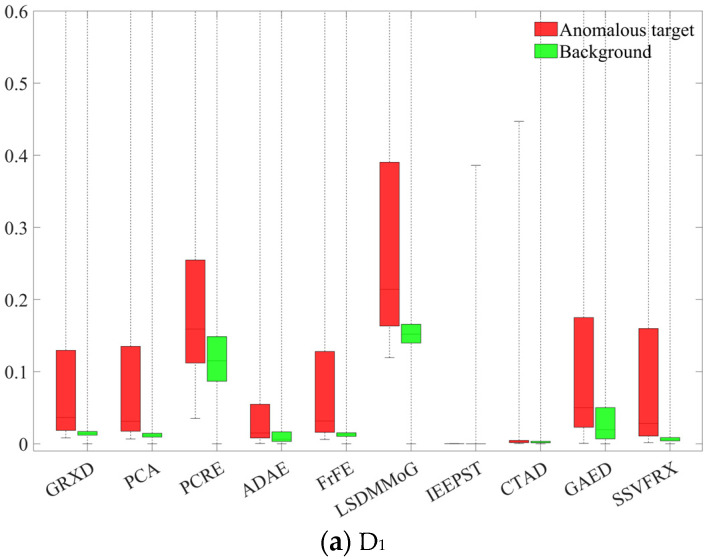
Comparison of SSA of different methods on different datasets.

DRI is a two-dimensional flat view that uses color depth to represent an anomaly. As shown in the legend in Figure 8, the value represents the probability that the sample is an anomaly. DRI contains spatial information that can be used to observe differences between various categories, including anomaly target and background. As can be seen from Figure 8, the contours between the anomaly target and the background are clearer, and more separable for SSVFRX than for the other comparison algorithms.

As shown by the running time in Table 10, the SSVFRX algorithm exhibits a higher computational cost compared with the other compared algorithms. This is mainly due to the fact that the algorithm is more sensitive to the data dimension as well as the complex structure containing two sets of deep learning networks. Therefore, the runtime increases significantly when dealing with the higher dimension dataset (D_2_).

## 4. Discussion

The hypothesis is that SSVFRX has a greater degree of difference between the background and anomaly target, which can lead to better detection accuracy. The experimental results show that this hypothesis is correct. By theoretical analysis, the superiority of the SSVFRX algorithm may be due to the following reasons.

First, through network training, features of the same type are more likely to be mapped to the same direction. Second, the similar neighborhoods of an anomaly target tend to differ to a greater extent from themselves. Third, the trained model tends to match the characteristics of most data, and the errors arising from a smaller number of anomaly targets account for a lower proportion of the back propagation.

The possible reasons for the superiority of SSVFRX are analyzed based on the following experimental results:

Firstly, the 3D ROC detection results (Table 3, Table 4, Table 5, Table 6, Table 7, Table 8 and Table 9 and Figure 4, Figure 5, Figure 6, Figure 7, Figure 8, Figure 9, Figure 10, Figure 11 and Figure 12) indicate that, in most cases, SSVFRX significantly improves both target detectability and background suppression for datasets D_4_, D_5_, and D_7_, which is relatively low compared with CTAD. A similar trend is observed in the SSA analysis (Figure 13). SSA shows that, except for D_4_, D_5_, and D_7_, SSVFRX enhances the separability of backgrounds and anomaly target more effectively. A possible reason for this is that CTAD is a local anomaly detection method, which has some advantages in highlighting anomalies when compared with the global detection algorithm of SSVFRX. However, both anomalies are relative to different backgrounds, and anomalies in the global scope do not necessarily belong to anomalies in the local scope, and vice versa. Therefore, CTAD exhibits relatively weaker background suppression capabilities, as evidenced by its lower performance compared with SSVFRX in datasets D_4_, D_5_, and D_7_ (Table 6, Table 7 and Table 9). This is further validated in the detection results in Figure 14. Comparing the CTAD and SSVFRX in Figure 14, it is clear that there is more false detection in the CTAD background, while the background is clearer in the SSVFRX. A possible reason for this is that the suppressed background in CTAD is not the global background. Thus, it is easy to produce a situation where the background is mistaken for an anomaly.

Second, it can be seen from the DRI (Figure 14) that SSVFRX obtains a clearer contour of the anomaly target, representing a better separation between the anomaly target and the background. It can be inferred that SSVFRX is able to increase the difference between the background and the anomaly target. 

**Figure 14 sensors-24-05664-f014:**
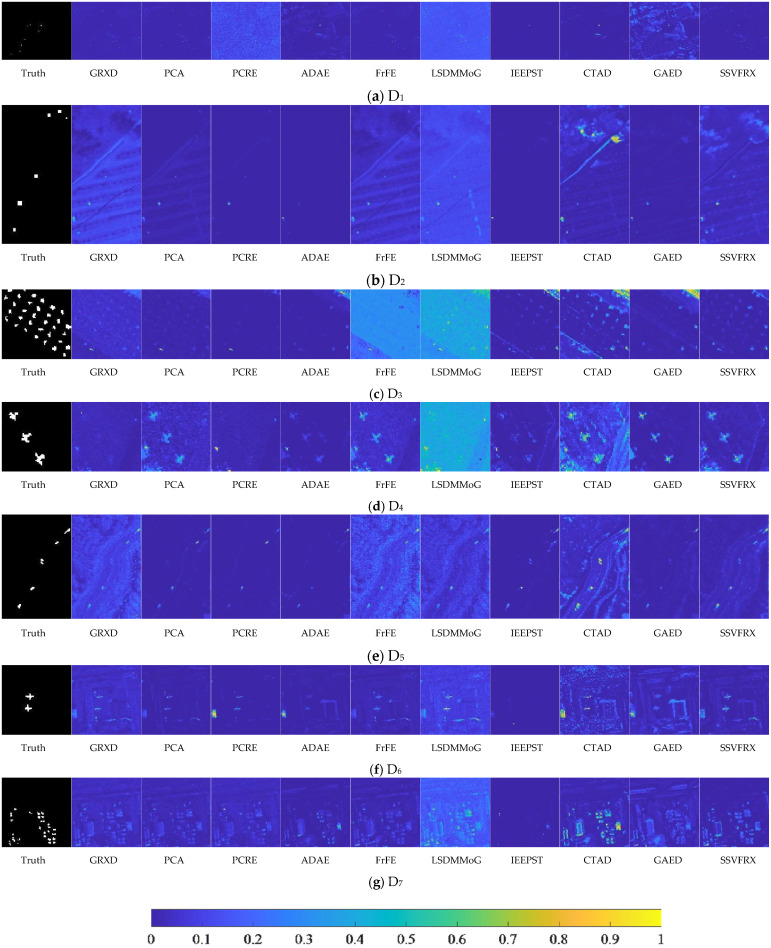
Detection results of different methods on different datasets.

There are also clearer contours in the background, but they are shallower than those of the anomaly targets. A possible reason for this is that SSVFRX makes different categories of samples map to the direction of their similar neighborhoods. One of the anomaly targets belongs to an isolated point, which makes it very different from its similar image elements. While the other categories of ground objects in the background have smaller differences from the similar image elements. This is the reason that SSVFRX is able to suppress the background better.

Third, the running time of the SSVFRX algorithm (Table 10) shows a relatively high computational cost. Nevertheless, it demonstrates significant advantages in key aspects such as detection accuracy, background suppression, and anomaly target highlighting. In practical applications, it is necessary to balance the algorithm’s performance and efficiency according to specific requirements. For scenarios where real-time processing is not critical but high detection accuracy is crucial, the advantages of SSVFRX may far outweigh its runtime drawbacks. Moreover, improvements can be made to reduce computational costs. First, dimensional reduction methods can be considered. For example, in a group of data sets, D_1_ and D_2_ are compared to show that, when the sample size is large, a lower dimension reduces the computational cost. Second, parallel computing or GPU acceleration techniques can be utilized to enhance the algorithm’s execution speed.

The SSVFRX algorithm shows significant advantages in anomalous target detection tasks, being able to improve both background and anomalous target separability and background suppression, especially in terms of background suppression. These advantages mainly stem from the efficient mapping of different classes to their similar samples. While local detection methods may have their advantages in some specific cases, global SSVFRX is more advantageous in terms of background suppression.

## 5. Conclusions

SSVFRX is capable of capturing rich anomaly and difference information, effectively distinguishing different types of features, and highlighting anomaly targets. Experiments have shown that SSVFRX is able to improve target and background separability and background suppressibility at the same time. The advantages of SSVFRX are mainly reflected in several aspects: the anomaly target is shown as an isolated point, and its similar features are different from the original features, SSVFRX can accurately capture such differences and improve the accuracy of anomaly detection. Meanwhile, SSVFRX maps the background to a similar direction to enhance the background suppression effect, which improves the detection capability and reduces the false alarm rate. However, there is still room for improvement in computational efficiency, such as optimizing network structures, developing high-dimensional data processing techniques, exploring optimal parameter configurations, and leveraging parallel computing or GPU acceleration. Through continuous optimization, the efficiency and performance of SSVFRX are improved, so that it can play a greater role in the field of hyperspectral anomaly detection.

## Figures and Tables

**Figure 2 sensors-24-05664-f002:**
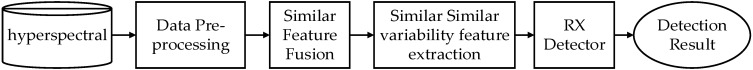
The overall flow chart.

**Figure 3 sensors-24-05664-f003:**
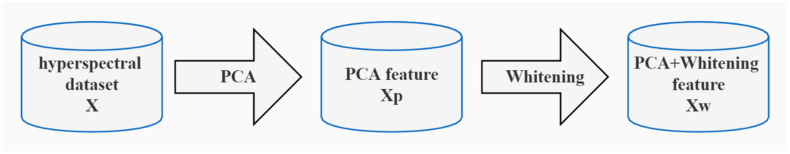
The flow chart of pre-processing.

**Figure 4 sensors-24-05664-f004:**
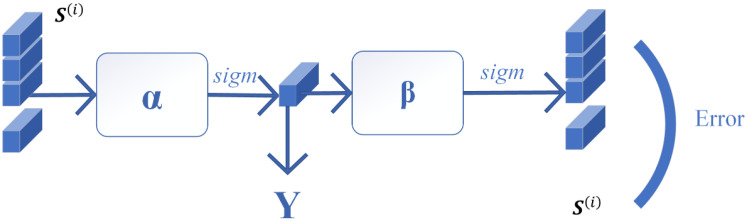
Similar feature fusion based on autoencoder.

**Table 2 sensors-24-05664-t002:** The relationship between parameter k and AUC.

Q	3	5	7	9	11
AUC	0.9414	0.9575	0.9603	0.9613	0.9583

**Table 10 sensors-24-05664-t010:** Running time comparison of different methods on different datasets.

Time	GRXD	PCA	PCRE	ADAE	FrFE	LSDM MoG	IEEPST	CTAD	GAED	SSVFRX
D_1_	0.37	1.03	4.14	464.02	99.54	46.60	1.9057	285.80	256.48	280.65
D_2_	0.76	12.54	104.63	2983.21	416.08	72.46	1.1473	127.72	69.08	7556.20
D_3_	0.07	7.38	74.82	195.52	7.80	3.59	0.1850	9.26	10.81	461.94
D_4_	0.03	2.44	12.44	79.78	4.41	1.56	0.1961	5.21	6.50	203.16
D_5_	0.24	2.49	28.83	938.90	121.58	26.65	1.8438	90.73	28.10	2355.64
D_6_	0.09	1.01	15.62	126.97	17.53	10.98	0.6469	23.78	22.48	1213.36
D_7_	0.10	1.12	7.12	133.86	18.87	14.79	0.6713	23.95	22.30	1219.95

## Data Availability

The datasets generated and/or analyzed during the current study are available from the corresponding author on reasonable request.

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
