# Peer review of "Hyperspectral Anomaly Detection Based on Spectral Similarity Variability Feature"

_sensors, 2024, doi:10.3390/s24175664_

Round 1

Reviewer 1 Report

Comments and Suggestions for Authors

The work uses a SSVF approach to enhance hyperspectral anomaly detection. Although not too much innovation, it is practical and useful. The authors compared with other existing approaches and prove it to be useful. However, the introduction part lacks of logicality to some degree. There is a phenomenon of literature stacking. The authors use too many reference []’, that is annoying. Readers want to a smoother reading experience.

Reviewer 2 Report

Comments and Suggestions for Authors

The paper proposes SSVF model for anomaly detection using HSI. The model was compared to several state of art methods. The experimental results reveal the outperformance of the model with several accuracy assessment metrics. Local RX, global RX, dual window RX, kernel RX, subspace RX, and the global RX with a uniform target detector are developed in anomaly detection and have been widely used. Revisions and recommendations have been proposed to the author to enhance the quality and presentation of the manuscript.

1.       Please correct the abstract section. Abstract should contain the main quantitative findings of your study.

2.       Proposed algorithm Section: Please note that there is a lake of consistency, It is proposed to give a little introduction of this section and to homogenise the nomenclature ‘pre-treatment’ , ‘pre-processing’ …

3.       Revise the second paragraph in the introduction section.

Within the introduction first paragraph, It is suggested to add recent and relevant citations in the classification, target-detection and super-resolution of HSI. See:

10.1109/JSTARS.2023.3242048

https://doi.org/10.1016/j.rsase.2024.101218

https://doi.org/10.1016/j.jag.2024.103816

https://doi.org/10.3390/rs14091973

4.       Your method uses the Reed and Xiaoli deterctor algorithm. I recommend to add -RX with your model name.

5.       Experimental results:  The data characteristics in Fig.5 and Table 1 should be moved to a previous section showing the used materials.

6.       Separability analysis reveals more accurate results with the benchmarking methods than the proposed method especially in D1, D2, D4. Provide more explanations of that.

The grammatical quality of the manuscript should be enhanced. Some points are as follows. However, please have the paper proofread by a native English speaker.

- Rewrite line 178: secondly should be second.

- Rewrite the line 193

- Rewrite the lines 208-209 properly

- Rewrite the paragraph in line 212.

7.       Table 1: The used datasets show a distinct GSD varying with 0.6, 3, 1, 7.1 meters. Provide a discussion of the effect of these various GSDs in the performance SSVF as well as the used methods.

8.       Discussion section: the spectral similarity variability feature (SSVF) method limitations should be highlighted and compared to other state of art methods. Provide the limitations and the future prospects of your research.

9.       Figure 3. Please correct ‘hyperspectral’ and provide a proper figure.

10.   Figure 8. Please remove the black outlines.

Comments on the Quality of English Language

English quality may need improvements.

Reviewer 3 Report

Comments and Suggestions for Authors

The article proposes Hyperspectral anomaly detection based on the Spectral Similarity Variability feature, but there are still the following issues that need to be addressed.

1、This paper has many formatting issues that need to be corrected.

(a) Proper nouns should appear in their complete form before using abbreviations, such as HAD, JGD, HSI, GFT, FrFD...... .

(b) In addition, please ensure proper citation when introducing other works, using the full name first if necessary, such as:

the research of HAD……(on page 2, line 52)

“The JGD model…… (on page 2, line 64)”

First, the AE network is……(on page 2, line 90)

(c) Why are there two different fonts used between lines 52 and 80 on page 2?

(d) Please ensure consistency throughout the entire paper: Reference [X] or In Ref. [X].

2、Please type setting the figures in the paper so that no figure spans across two pages.

3、(a) As described in section 3.1 'Experiment Data Description,' is the data compared in this paper only a small portion of the overall dataset? Could you compare the full dataset? If not, please explain why. (b) Could you use 1-2 public datasets for anomaly detection to demonstrate the effectiveness of the algorithm?

4、The comparison algorithms are outdated. Please update with 3-4 works from the years 2022-2024.

5、As described in section 3.3 ' Parameter selection,' are α,β, and T the only parameters affecting the performance of the proposed method? If there are other hyperparameters, please explain in more detail why these three parameters are the main ones.

6. The requirement to cite the source of evaluation indicators in academic literature.

7. The reference formatting does not comply with the requirements of the Sensors journal. Please revise accordingly, ensuring adherence to the journal's guidelines.、

8. To plot an ROC curve with the x-axis ranging from 0 to 1

9. Please add more evaluation criteria; the current indicators are too few. The authors can refer to the following literature:

[1] Fan G , Ma Y , Mei X ,et al.Hyperspectral Anomaly Detection With Robust Graph Autoencoders[J].IEEE Transactions on Geoscience and Remote Sensing, 2022, 60.DOI:10.1109/TGRS.2021.3097097.

10. The true-value of the data, I suggest using a binary image for better visibility and observation.

Comments on the Quality of English Language

Moderate editing of English language required

Round 2

Reviewer 3 Report

Comments and Suggestions for Authors

After reviewing the revised manuscript, I still have the following issues that need to be addressed by the author

1. It has been shown in many references that the AUC value of (PD,PF) cannot be used to evaluate (BS). In addition, the false alarmed is generally referred to FA or PF not FD which is msileading. This reviewer was surprised by the fact that the authors were not aware of recent publications using 3-D ROC curves.

2. Although AUC is a quamtitative measure, it cannot address target detectability and BS this is because PD and PF are determenied by the same threshold and cannot work alone independently. As a result, the experimental results are not convincing.

3. The comparison algorithms do not include any newer algorithms from 2023 or 2024.

4. The format of the references needs further standardization, including consistency in capitalization.

5. I suggest adding the ground truth images to Figure 8.

6. I suggest that the author references some recently published relevant papers in the journal Sensors.

Comments on the Quality of English Language

Minor
